# Laser-Induced Interdigital Structured Graphene Electrodes Based Flexible Micro-Supercapacitor for Efficient Peak Energy Storage

**DOI:** 10.3390/molecules27010329

**Published:** 2022-01-05

**Authors:** Apurba Ray, Jenny Roth, Bilge Saruhan

**Affiliations:** German Aerospace Center (DLR), Department of High-Temperature and Functional Coatings, Institute of Materials Research, 51147 Cologne, Germany; apurba.ray@dlr.de (A.R.); Jenny.vonWnuckLipinski@dlr.de (J.R.)

**Keywords:** CO_2_-laser-induced graphene, ionic electrolyte, micro-supercapacitor

## Abstract

The rapidly developing demand for lightweight portable electronics has accelerated advanced research on self-powered microsystems (SPMs) for peak power energy storage (ESs). In recent years, there has been, in this regard, a huge research interest in micro-supercapacitors for microelectronics application over micro-batteries due to their advantages of fast charge–discharge rate, high power density and long cycle-life. In this work, the optimization and fabrication of micro-supercapacitors (MSCs) by means of laser-induced interdigital structured graphene electrodes (LIG) has been reported. The flexible and scalable MSCs are fabricated by CO_2_-laser structuring of polyimide-based Kapton ^®^ HN foils at ambient temperature yielding interdigital LIG-electrodes and using polymer gel electrolyte (PGE) produced by polypropylene carbonate (PPC) embedded ionic liquid of 1-ethyl-3-methyl-imidazolium-trifluoromethansulphonate [EMIM][OTf]. This MSC exhibits a wide stable potential window up to 2.0 V, offering an areal capacitance of 1.75 mF/cm^2^ at a scan rate of 5.0 mV/s resulting in an energy density (E_a_) of 0.256 µWh/cm^2^ @ 0.03 mA/cm^2^ and power density (P_a_) of 0.11 mW/cm^2^ @0.1 mA/cm^2^. Overall electrochemical performance of this LIG/PGE-MSC is rounded with a good cyclic stability up to 10,000 cycles demonstrating its potential in terms of peak energy storage ability compared to the current thin film micro-supercapacitors.

## 1. Introduction

Recently, the rapid growing development and demand of miniaturized portable and wearable electronics has expressively amplified the importance for lightweight, stretchable, microscale and efficient power storage systems [1,2,3]. Modern life is also becoming more expedient day by day by the extensively utilization of fast remote-control smart devices, gadgets and recent development of internet-of-things (IoT) systems [4,5]. An IoT system needs lot of devices and sensors to be useful in large scale industrial purposes. Sustainable energy harvesting technologies such as solar-cells, thermal or mechanical energy, wind power, piezoelectric or triboelectric nanogenerators, etc., are widely investigated as power resources for the sensors and devices of IoT [6,7]. However, the instability and intermittence of these power resources urgently require the development of high-performance energy storage systems (ESSs) well-matched with the present demand. Accordingly, different types of miniaturized electrochemical energy storage systems (ESS) such as micro-batteries (MBs), micro-supercapacitors (MSCs), etc., have been widely explored over recent years to develop efficient EES systems [8,9]. Up to date, market available MBs have exhibited promising performance as miniaturized EES systems, but it is still challenging to overcome their intrinsic limitations including low power density, low lifetime, etc., which hinder their practical application [10,11]. Alternatively, electrochemical capacitors or supercapacitors (SCs) have drawn lot of interest as a novel ESSs over MBs due to promising advantages such as high-power density, fast charge–discharge rates, long cycle life (>10,000) and ease of integration with various electronic components. Recent research has found that the MSCs are particularly attractive as one of the most competitive high-power sources for future miniaturized IoT technologies due to high power density, small size, controllable patterning, large scale on-chip integration ability and long cycle life [4,12]. Usually, MSCs can be fabricated by different ways such as stacked thin films of the cell components or planar interdigitated methods, etc., and it has also been observed that the nanostructured functional materials such as polymer films, graphene, carbon-based nanocomposites electrodes based thin-film flat MSCs can be advanced components for modern integrated circuits. The charge storage mechanism via capacitive ion adsorption/desorption in porous carbon electrodes based MSC can offer high power density and long cycle life which can be characterized by rectangular shape of cyclic voltammograms (CV) curves and linear nature of galvanostatic charge–discharge (GCD) curve at a constant current. Pseudocapacitive electrode materials based MSCs also exhibit similar features due to fast surface redox reactions of the cation in the active electrode material. There is no such phase change generally observed in redox active electrode materials during long cycle life which signifies the capacitive behavior of the MSC [7,13]. 

However, the up-to-date literature displays that the MSCs still have deficits in respect to simultaneous high specific capacitance and power density due to limited ionic conductivity of the applied solid electrolytes. One of the major issues that stand before the current practical application of MSC is its internal short circuit problem [14,15]. One promising way to overcome this challenge is to increase the electrode thickness by loading of more active electrode mass with a suitable interelectrode gap. Consequently, this method can enhance the charge transfer resistance but leads to lower power density. Thus, the simultaneous obtainment of high areal energy density as well as high power density at a MSC becomes a major challenge [12,16]. This issue can be encountered by coupling the nano-porous carbon electrodes with suitable electrolytes that offer a wide stable electrochemical potential window to accomplish superior MSCs exhibiting higher capacitance values, higher energy density and long cycle stability [10,11,17]. 

Hence, this research work targets to emphasize the fabrication of LIG-based EDLC electrodes at ambient temperature for use in flexible and scalable micro-supercapacitor. Further ambition is to display the significant impact factors for achievement of improved MSC performance through embedded ionic liquid electrolyte and optimized ion transport pathways via the increased electrode porosity resulting in a wider potential window. In this context, we report the optimization and fabrication of MSCs based on laser-induced interdigital structured graphene electrodes fabricated by CO_2_-laser structuring of polyimide-based Kapton foils. Furthermore, PPC polymer matrix embedded [EMIM][OTf] ionic liquid is used yielding a polymer gel electrolyte (PGE). The detailed electrochemical device performance of this MSC have been carried out by cyclic voltammogram (CV), galvanostatic charge–discharge (GCD), electrochemical impedance spectroscopy (EIS) measurements and determination of cycle life by means of two-electrode measurements that are explained below. 

## 2. Results and Discussion

The SEM images have been recorded at different magnifications of the samples at different condition to determine the change of surface morphologies of the PI-film before and after processing of irradiation with CO_2_-Laser. Appendix A represents the influence of surface etching using KOH-solution and it shows that the commercial PI-foil has a smooth surface except for a few minor visible scratches (Appendix A). Nonuniform and agglomerated formation of KOH crystals (Appendix A) is observed on the PI- surface due to the etching and drying process with 1 M KOH solution, but after etching and several times washing with DI- water for KOH removal, a clean and deeper indentation (Appendix A) on the surface is observed. However, no noticeable reduction in PI film thickness is detected in the cross sections after the etching process. The top view SEM-image (Figure 1a) of the successful conversion of PI-foil surface into LIG exhibits a prominent porosity distribution on the surface of PI-foil after quite sharp and straight irradiation with Laser. The pore distribution on the electrode surface plays a vital role for electrochemical charge storage performance of a device [18,19]. Larger number of pores on the electrode surface enhances the infiltration of the electrolyte ions within the pores leading to the increase of capacitance value as well as energy density of the device [17,20]. The EDX mapping area (Figure 1b) with different colors also represents the oxygen rich (green color) and carbon rich (red color) areas of the LIG converted PI-foil. High intensity coloring represents a high atomic percentage of oxygen (O) and carbon (C) elements. As the investigated domains, PI and LIG are at different heights, a higher excitation voltage of 15 kV was used. The SEM image (Figure 2a) shows the morphology of the LIG converted of from PI-foils with a laser power of 6 W.

The conversion or yield of graphene layer thickness obtained through laser irradiation depends mainly on the laser power as well as surface treatment of the PI-foil. This is displayed by applying different laser powers and use of untreated and KOH surface treated foils. By starting with an untreated (no pre-treatment (No PT)) PI-foil of 125 µm thickness and a laser power of 4.8 W, approx. 18–30 µm polymer was converted into porous carbon-based layer. The thickness of this resulting porous layer varied depending on the applied process parameters between 50–94 µm. The increase of laser power from 4.8 W to 5.6 W has also increased the converted PI-foil thickness from 18 to 31 µm. A further increase of the laser power to 6 W, the penetration depth of the laser decreased to 20 µm, indicating a laser power of 5.6 W yields maximum conversion of the untreated PI-foil surface. On the other hand, a surface pre-treatment of the PI-foil using KOH (KOH-PT) yields a significant growth of the LIG layer at higher laser powers (5.2–6 W). The maximum LIG layer thickness (94 µm) has been achieved with 6 W and on the KOH pretreated PI-foils. These resulted also in better layer morphologies as well as higher yields. While the PI surfaces irradiated without any pretreatment deviated slightly from a linear increase. However, over 6 W exceeding increase of laser power (e.g., 6.4 and 8 W) on KOH-PT foil resulted in a largely deviating layer thickness and the wave-like surface topology.

Overall, it was observed that the surface porosity of the film is significantly different from the center of the film, i.e., when approaching towards the PI-surface, the pore diameters become more homogeneous and smaller. Approximately the porous LIG resulted through the conversion of Kapton foil has a thickness of varying between 50 and 94 µm. Conclusively, this micro-branched and/or columnar microstructure (Figure 2a, inset) of these porous LIG electrodes are beneficiary for electrolyte ions conduction providing short diffusion path.

The X-ray diffraction (XRD) spectroscopy measurement has also been performed to analyze the conversion of PI-foil to LIG. The XRD figures (Figure 2b) of bare PI-foil and LIG clearly demonstrate the successful conversion of PI-foil substrate to LIG-electrode. The broad polymer peaks at 2θ(degree) = 22.3, 26.4 and 36.0 of PI-foil have almost disappeared after laser treatment. The significant shift of the LIG samples towards a smaller angle 2θ(degree) = 26.0 and 42.7 results in larger interplanar spacing of about 3.6 Å, which may be due to the introduction of a several number of oxygen (O)-containing groups on each layer of LIG and conversion of LIG samples corresponding to the planes (002) and (100), respectively [21,22,23,24].

The electrochemical performance of as-prepared flexible LIGs//LIGs symmetric micro-supercapacitor (SMSC) has been firstly studied by cyclic voltammetry (CV) measurements (Figure 3a) at different scan rates from 5 mV/s to 100 mV/s. The CV curves for this SMSC represents almost typical rectangular in shape within a wide stable electrochemical potential window from 0.0 to 2.0 V, which reveals an ideal capacitive behavior of this LIG-electrodes [25]. The CV curves even at high scan rate of 100 mV/s continue to display the rectangular shape without any degradation, signifying a fast charge transfer capability within the electrode materials and relatively low equivalent series resistance (ESR) of this MSC. This flexible MSC also exhibits wide potential window up to 2.0 V with better rate performance compared to other interdigital structural MSCs [4,26,27,28]. The linear increase of capacitive current with increase of scan rates from 5 mV/s to 100 mV/s for this LIG-MSC is also observed, which suggests its fast electrolyte ions diffusion, fast charge transport and high-power output ability [29,30]. To the best of our knowledge, this LIG-MSC shows highest potential window up to 2.0 V with PPC polymer embedded [EMIM][OTf] ionic liquids-based electrolyte among all reported MSCs based on other graphene and carbon electrodes [31,32,33,34]. The capacitance value by means of total amount of charge stored in this LIG-MSC at a fixed scan rate can be determined from the total area covered by the CV curve. The areal capacitance (C_A_) and specific capacitance (C_m_) are calculated from these CV curves using Equations (1) and (5), respectively. The maximum areal capacitance (C_A_) of 1.75 mF/cm^2^ and specific capacitance (C_m_) of 629.5 mF/g, respectively is obtained at 5 mV/s scan rate. The variation of areal capacitance (C_A_) vs. scan rate (Figure 3b) represents that the capacitance value decreased with increasing scan rate due to limitation of ions movement from [EMIM][OTf]: PPC electrolyte to LIG-electrodes. At lower scan rates, the electrolyte ions [EMIM]^+^/ [OTf]^−^ from electrolyte can take longer time to infiltrate within the electrode pores and thus, can access into the maximum inner and outer surface of the LIG-electrodes [35,36,37]. Therefore, the number of accumulated charges increase in the electrode/electrolyte interface, leading to the increase of capacitance value. Consequently, the mobility of charges per unit time increases at higher scan rates resulting in the charges access solely at the outer surface of the electrodes. Hence, the capacitance value decreases due to a smaller number of charge accumulation on the electrode/electrolyte interface [38,39,40]. On the other hand, these LIG-electrodes can also provide larger specific surface area and more electrochemical active surface sites for fully accessibility of electrolyte ions via adsorption/desorption to significantly enhance the charge capacitive performance of the MSC device.

In order to investigate the charge/discharge behavior of this LIG-MSC, the galvanostatic charge/discharge (GCD) studies at different current densities (0.03, 0.05, 0.07 and 0.10 mA/cm^2^) are performed. The GCD curves (Figure 4a) for all currents are symmetric in nature implying typical non-faradic or EDLC charge storage mechanism of these LIG-electrodes. Low voltage drops (IR-drop) at the starting of discharge curves even at higher currents signifies good reversibility, low electrolyte resistance as well as good electrical contact between electrodes and PI-current collector. However, the porous morphology of this LIG-electrode plays a significant role to accelerate the movement of [EMIM]^+^ cations and [OTf]^−^ anions through electrode channels inside the pores indicating good capacitive behavior of this MSC. The areal capacitance (C_A_) and specific capacitance (C_m_) are also calculated from GCD curves at different current densities using Equations (2) and (6). The maximum areal capacitance (C_A_) of 0.461 mF/cm^2^ and specific capacitance (C_m_) of 166.1 mF/g are obtained at 0.03 mA/cm^2^ current density (Appendix A). Areal capacitance (C_A_) vs. current density curve (Figure 4b) reveals that the capacitance value decreased with increase of current densities due to limited mobility and accessibility of the electrolyte ions at higher current densities as discussed earlier [12,41]. The Ragone plot drawn by means of areal energy density (E_A_) vs. areal power density (P_A_) of this LIG- electrode based MSC device (Figure 5a) exhibits that maximum areal energy density of 0.256 µWh/cm^2^ @0.03 mA/cm^2^ and areal power density of 0.11 mW/cm^2^ @0.10 mA/cm^2^, are achievable (Appendix A). The overall electrochemical performance of this LIG-MSC comparison with other reported work has been presented in Table 1.

The long cycle life study of this as prepared LIG-MSC device is also performed up to 10,000 cycles at a constant current of 0.1 mA/cm^2^. The cycle performance of this MSC device (Figure 5b) shows good retention of 98.12% over 10,000 cycles. The initial capacitance value starts to increase with increasing cycle number up to 500 cycles, which may be due to the electrochemical activation effect of the LIG-electrode materials and then begins to decrease slightly and maintain almost same values [25,46]. It has been observed that the combination of these laser-induced interdigital structured graphene electrodes and PPC polymer embedded [EMIM][OTf] ionic electrolytes play a significant role for improvement of electrochemical performance and long cycle stability of this flexible MSC device. Overall, the good electrochemical performance and good cycle stability suggests that this laser-induced interdigital structured graphene electrodes based flexible micro-supercapacitor could be one of the promising energy storage systems for efficient peak energy storage in future.

The electrochemical impedance spectroscopy (EIS) study is performed over the applied frequency range from 100 kHz to 0.01 Hz at AC perturbation amplitude of 10 mV to understand the charge storage behavior of this device. The EIS plot (Figure 6) consists of two distinct portions, such as one depressed semicircle portion in high frequency region (Appendix A) corresponding to charge transfer resistance (R_ct_) and other inclined straight-line portion in low frequency region corresponding to the charge diffusion resistance, called Warburg impedance (W_0_), respectively. The initial intercept at the Z’-axis in the Nyquist plot represents the equivalent series resistance (ESR) or contact resistance (R_s_) due to the combination of electrolyte resistance and intrinsic resistance of LIG-electrode materials [42,44]. To get the equivalent circuit model, this EIS plot is well fitted using EC-Lab software as shown in Figure 6 (inset). Among three parts of the equivalent circuit, the first part provides the value of equivalent series resistance (ESR) or Rs of 752.1 Ω. The second part by means of parallel combination of constant phase element (CPE1), Warburg impedance (W_0_) and charge transfer resistance (R_ct_), respectively represents quite common charge transfer phenomenon of porous graphene electrodes. The relatively higher R_ct_ of 348 Ω signifies the lower infiltration of electrolyte ions or higher ionic transfer resistance effect within the smaller pores inside the electrode active sites [9,11,17]. The third portion of the circuit model CPE2 represents the electrochemical activity or kinetics occurred at the electrode/electrolyte interface of LIG-electrodes. The goodness of fitting factor χ^2^) of 0.00161 proposed good fitting of this EIS plot.

## 3. Materials and Methods

### 3.1. Materials

The 1-ethyl-3-methyl-imidazolium-trifluoromethansulphonate [EMIM][OTf] (99%, M = 260.24 g mol^−1^, 𝜌 = 1.39 g cm^−3^, IoLiTec, Heilbronn, Germany), polypropylene carbonate (PPC) (94.5%, Mn=50,000 g mol^−1^, Sigma Aldrich, Taufkirchen, Germany), acetonitrile (99%, Sigma Aldrich), polyimide-based Kapton foils (Kapton^®^ HN 125 ± 13 µm, CMC Kleebetechnik, Germany), Isopropyl alcohol (IPA) (99.5%, ACROS Organics) and potassium hydroxide (KOH) (85%, Riedel-de-Häen, Germany) were used for this experiment. All chemicals were of analytical grade and used without further purification.

### 3.2. Fabrication of CO_2_-Laser-Induced Interdigital Structured Graphene Electrodes

A commercial polyimide (PI)-based Kapton^®^ HN foil (hereafter given as Kapton Foil) with a thickness of 125 μm was used as a flexible substrate and as the carbon precursor for the production of graphene layers. Prior to the laser irradiation with CO_2_-Laser, the Kapton foil surface was pretreated with a 1M KOH solution. For that, the KOH solution was drop-casted onto the Kapton foil surface which was then dried in a vacuum desiccator for overnight under reduced pressure. Following this, the Kapton foil was attached onto an aluminum oxide ceramic plate by means of a double-sided Kapton tape to fix and stabilize the Kapton foil for further irradiation processing with CO_2_-Laser (supplied by VE-VOR, Germany). The irradiated Kapton foil was washed with isopropanol solution and dried in a vacuum desiccator over silica gel pellets at reduced pressure. After this pretreatment and stabilization, the Kapton foil was scanned in a continuous wave (CW) CO_2_-Laser engraving machine with λ = 10.6 µm in the geometry shown in Figure 7. The dimension of such fabricated MSC-devices was 24 × 10.5 mm. The interdigital Graphene electrodes displayed a hair comb formation which had 12 branches of 7.5 × 0.8 mm equally distributed on each bar of the comb (Appendix A). The laser scanning speed was 8 mm s^−1^ and had a laser power of 15 % of the maximum power being equivalent to 6.0 W. The distance between the lens and substrate was 55 mm. The manufacturing of the cells was performed at room temperature and in ambient air.

### 3.3. Preparation of Polymer Gel Electrolyte (PGE)

To prepare the polypropylene carbonate (PPC) polymer embedded 1-Ethyl-3-methyl-imidazolium-trifluoromethansulphonate [EMIM][OTf] ionic liquid-based polymer gel electrolyte (PGE), PPC was added into acetonitrile (AN) solvent in the approximate ratio of 1 gm PPC in 5 mL AN. The solution was first stirred at 90 °C for 30 min until the PPC was completely dissolved. Then half amount of PPC, the [EMIM][OTf] ionic liquid was added in the solution to maintain the weight ratio of [EMIM][OTf]: PPC = 1:2 or 0.5. This polymer solution was then stirred at 90 °C for another 30 min to make the solution homogeneous. To make the electrolyte viscous, this electrolyte was stirred for another 2 h at room temperature and finally used it for MSC device fabrication. The preparation of this PGE has been followed from one of our previous works [47].

### 3.4. Fabrication of In-Plane Micro-Supercapacitor (IMSC) Devices

The laser-induced interdigital structured graphene electrodes to prepare LIGs//LIGs symmetric IMSC is realized as shown schematically in Figure 8. Before the components were assembled to fabricate the MSC, the laser-induced graphene (LIG) printed electrodes were washed several times using deionized water and isopropanol solution. The interdigital electrodes were used as stability-providing components such as electrodes as well as current collectors. For electrical connection, single sided copper SEM adhesive tape was used as shown in Figure 8. Approximately, 50 µL of the [EMIM][OTf]: PPC electrolyte was dropped on the active electrode surface and dried overnight in a vacuum desiccator before further electrochemical characterization is performed.

### 3.5. Characterizations

To investigate the morphology of the samples, the field-emission scanning electron microscopy (FE-SEM) images were produced using an Ultra 55 from Zeiss (Jena, Germany). The samples were glued to a standard aluminum sample carrier by means of conductive carbon adhesive pad. The SE in lens detector with an accelerating voltage of 5 kV and a working distance of approximately 8.5 mm was used for this measurement. Depending on requirements, images were taken with magnifications between 115 and 50 k. The X-ray diffractograms (XRD) were taken to investigate the material conversion from polyimide (PI) to graphene-like structure and to confirm using a Bruker D8 Advance with a Göbel mirror employing Cu K_𝛼_ radiation with λ = 1.54 Å in the 2θ range of 15–60°. The scan type was “Coupled Two Theta/Theta Bragg–Brentano” at a scan speed 1 s/step and the change in angle was 0.05°. Due to the small amount of the sample the silicon single crystal holder was used as sample holder. The phase analysis was performed at the DIFFRAC.EVA software from Bruker. Energy dispersive X-ray (EDX) spectroscopy measurements were also carried out to determine the elemental composition of the samples on surfaces using Aztec from Oxford Instruments (Wiesbaden, Germany). An acceleration voltage of 15 kV was chosen for the measurement and the working distance remained the same level.

The electrochemical characterization of the LIG- based flexible micro-supercapacitors (MSCs) with a geometric electrode area of 0.72 cm^2^ and active mass loading of 2.0 mg were conducted on a Gamry Reference 3000 (provided by C3 Prozess- und Analysentechnik GmbH, Haar b. Munich, Germany) workstation using two-electrodes configuration by cyclic voltammetry (CV), electrochemical impedance spectroscopy (EIS) and galvanostatic charge–discharge (GCD) measurements. For better shielding of unwanted external effects, all the electrochemical tests were performed in the Gamry calibration shield, which itself acts as a Faraday cage. Gamry’s data acquisition program Framework^TM^ was used for parameter in-put and visualization of the running measurements. The areal capacitance (C_A_, mF/cm^2^) and specific capacitance (C_m_, mF/g) of a supercapacitor can be calculated by CV or GCD techniques using Equations (1) and (2) or (5) and (6). The areal energy density (E_A_, μWh/cm^2^) or Sp. Energy density (E_m_, Wh/kg) and areal power density (P_A_, mW/cm^2^) or or Sp. Power density (P_m_, W/kg) can also be obtained by Equation (3) and (4) or (7) and (8) bellow, [25,31,41,48].
(1)$$C_{A}=\frac{\int_{V_{a}}^{V_{c}}i\;\left(V\right)\;dV}{2\times A\times \vartheta \times \Delta V}$$
(2)$$C_{A}=\frac{I\times \Delta t}{A\times \Delta V}$$
(3)$$E_{A}=\frac{C_{A}\times \Delta V^{2}}{7200}$$
(4)$$P_{A}=\frac{E_{A}\times 3600}{\Delta t}$$
(5)$$C_{m}=\frac{\int_{V_{a}}^{V_{c}}i\;\left(V\right)\;dV}{2\times m\times \vartheta \times \Delta V}$$
(6)$$C_{m}=\frac{I\times \Delta t}{m\times \Delta V}$$
(7)$$E_{m}=\frac{C_{m}\times \Delta V^{2}}{7.2}$$
(8)$$P_{m}=\frac{E_{m}\times 3600}{\Delta t}$$
where, $\int_{V_{a}}^{V_{c}}i\;\left(V\right)\;dV$ is the area under the CV curve at a fixed scan rate (*ϑ*) (V/s), *A* is the geometric electrode area (cm^2^), *m* is the active electrode mass (g), Δ*V* = (*V_c_* − *V_a_*) = voltage window (V), Δ*t* is the discharge time (s).

## 4. Conclusions

In summary, the fabrication of laser-induced graphene (LIG)-interdigital porous flexible electrodes using facile CO_2_-laser structuring of polyimide-based Kapton foils are demonstrated. Structural, morphological and compositional analysis have been performed using XRD, SEM and EDX studies. The micro-supercapacitor (MSC) devices are also fabricated using these LIG-electrodes and polypropylene carbonate (PPC) embedded 1-Ethyl-3-methyl-imidazolium-trifluoromethansulphonate [EMIM][OTf] polymer gel electrolyte (PGE). The MSC shows excellent supercapacitive performance in a wide potential window up to 2.0 V compared to reported graphene and carbon based MSC. This device offers maximum areal capacitance of 1.75 mF/cm^2^ at 5 mV/s scan rate measured from CV measurements. This MSC device also delivers 0.256 µWh/cm^2^ areal energy density (E_A_) at 0.03 mA/cm^2^ and 0.11 mW/cm^2^ areal power density (P_A_) at 0.10 mA/cm^2^. The overall electrochemical performances of this LIG- interdigital MSCs are fabricated by CO_2_-laser structuring of PI-based Kapton foils are quite impressive for future practical application as flexible micro-supercapacitor. This LIG-porous electrode, fabrication scheme, the PPC: [EMIM][OTf] polymer gel electrolyte (PGE) and the device could be a suitable promising applicability in the field of flexible micro-supercapacitor for efficient peak energy storage.

## Figures and Tables

**Figure 1 molecules-27-00329-f001:**
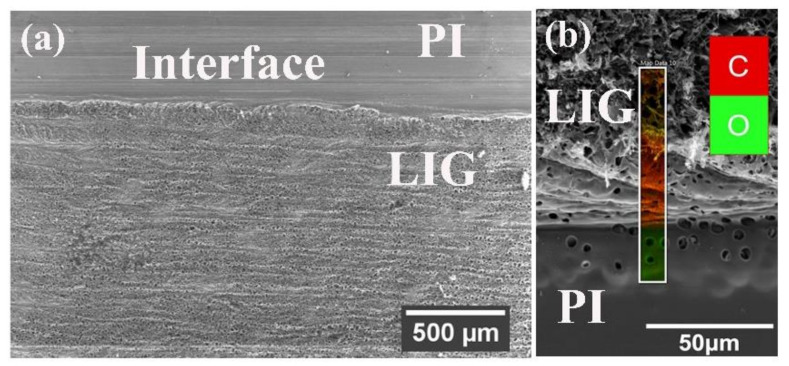
PI-foil converted to LIG (**a**) top view and (**b**) EDX mapping of the interface.

**Figure 2 molecules-27-00329-f002:**
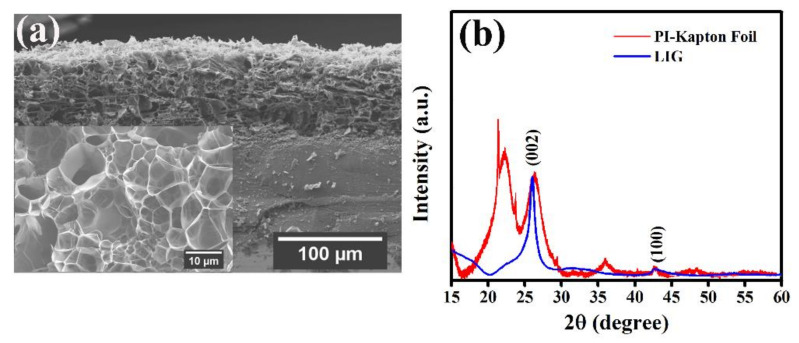
(**a**) Cross-section SEM images with porous structure view (inset) and (**b**) XRD plots of PI-Kapton foil and after conversion to LIG.

**Figure 3 molecules-27-00329-f003:**
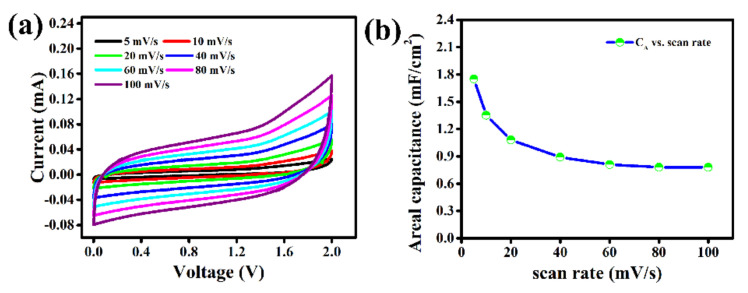
(**a**) Cyclic voltammograms at different scan rates and (**b**) areal capacitance vs. scan rate plot of LIG–MSC.

**Figure 4 molecules-27-00329-f004:**
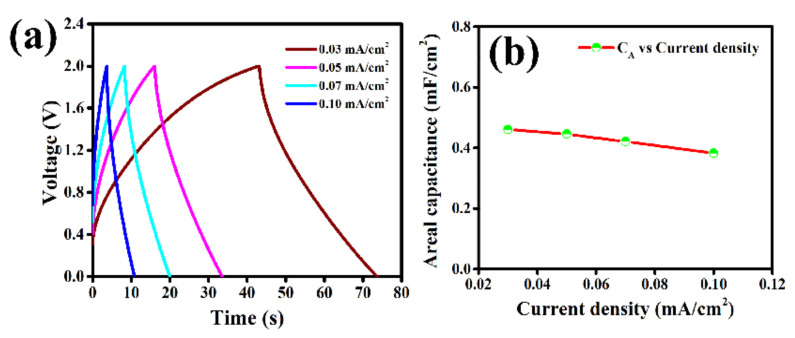
(**a**) Galvanostatic charge/discharge curves at different current density and (**b**) areal capacitance vs current density plot of LIG–MSC.

**Figure 5 molecules-27-00329-f005:**
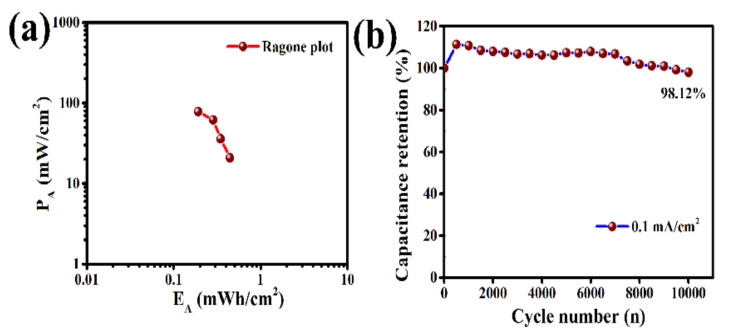
(**a**) Ragone plot and (**b**) capacitance retention vs. cycle number plot of LIG–MSC.

**Figure 6 molecules-27-00329-f006:**
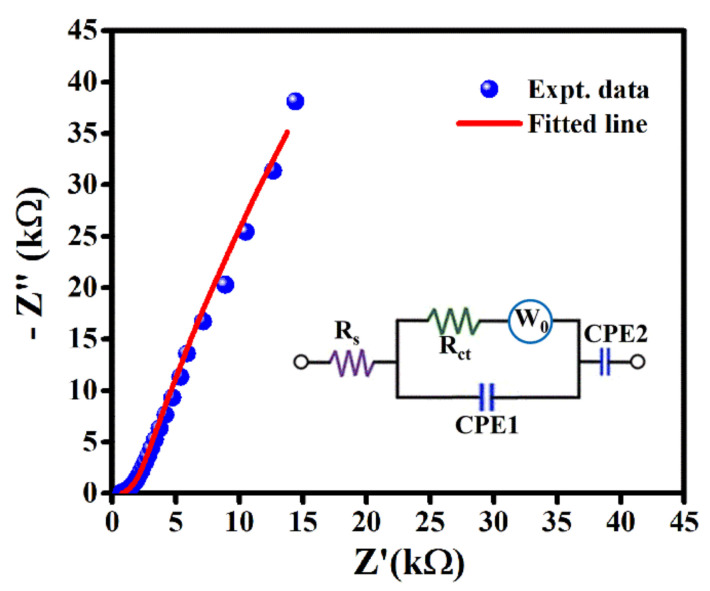
EIS plot and equivalent circuit model (inset) of this LIG–MSC.

**Figure 7 molecules-27-00329-f007:**
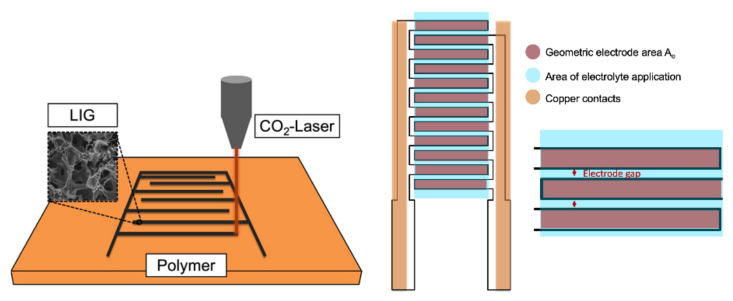
Schematic sketch of the converting PI-foil into porous LIG and sketch of an In-plane micro-supercapacitor (IMSC).

**Figure 8 molecules-27-00329-f008:**
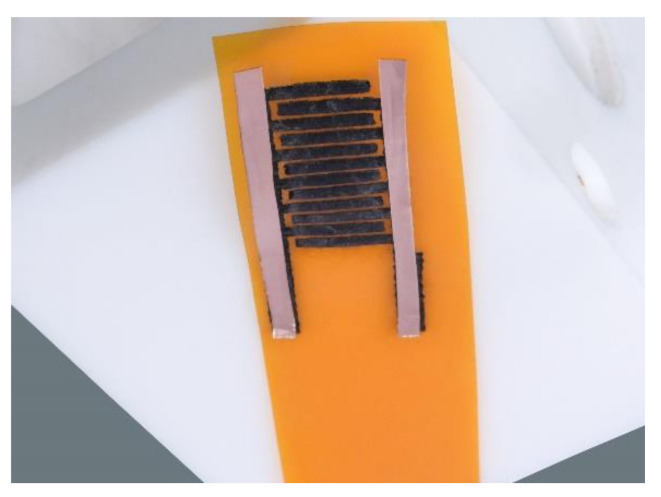
Fabrication of porous LIG -electrodes based in-plane micro-supercapacitor (IMSC).

**Table 1 molecules-27-00329-t001:** Some typical MSC device performance selected from literature.

Materials	Synthesis Method	Type of SC	Electrolyte	Capacitance	Energy Density	Power Density	Potential Window	Ref.
Ink-carbon fiber	coated by ink	SSC	LiCl-PVA	4.3 mF/cm^2^	34 µW h/cm^2^	0.38 μW/ cm^2^	0.8 V	[42]
rGO	“in-plane” fabrication approach	ASC	PVA	<0.4 mF/cm^2^	0.01 μWh /cm^2^	9 μW /cm^2^	1.0 V	[34]
Graphene	electrodeposition	ASC	0.5 M Na_2_ SO_4_	0.5 mF/cm^2^	0.07 μW h /cm^2^	7.5 μW /cm^2^	0.9 V	[43]
Mushroom derived carbon (MDC)	Scalable laser scribing technique	SSC	PVA-H_2_ SO_4_	9 mF/cm^2^	1.8 mWh/ cm^3^	720 mW/ cm^3^	1.0 V	[31]
Carbon	laser-induced MOF-derived	SSC	1 M H_2_SO_4_	5.02 mF/cm^2^	-	-	1.0 V	[44]
Graphene/CNTs	in situ	SSC	1 M Na_2_ SO_4_ IL	2.16 mF/cm^2^	0.16 mWh /cm^3^	115 W/ cm^3^	1.0 V	[45]
**Laser-Induced Graphene (LIG)**	**Laser-induced process**	**SSC**	**[EMIM][OTf]:PPC = 0.5**	**1.75 mF/cm^2^**	**0.256 µWh/cm^2^**	**0.11 mW/cm^2^**	**2.0 V**	**This work**

## Data Availability

The data presented in this work are available in the article and Appendix A.

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
