# Peer review of "Laser-Induced Interdigital Structured Graphene Electrodes Based Flexible Micro-Supercapacitor for Efficient Peak Energy Storage"

_molecules, 2022, doi:10.3390/molecules27010329_

Round 1
Reviewer 1 Report
Daer Authors
The paper is well conceived and the results are interesting. I recommend two changes to improve the paper:
1) Improve figure 1 adding a scheme of the "polymer sandwich" to better explain the device struture. In section 2.2 the authors first speak about PI, then kapton, then double side kapton tape, then of kapton HN on Alumina substrates. The stack structure remains unclear.
2) Measure and report the value of the electrical resistance of a single LIG strip, i.e. the surface resistance of the LIG film. This value should be correlated to the impedance plot in figure 8, at low frequencies.
Regards
Author Response
Reviewer 1.
Reviewer’s comments:
The paper is well conceived and the results are interesting. I recommend two changes to improve the paper:
1) Improve figure 1 adding a scheme of the "polymer sandwich" to better explain the device structure. In section 2.2 the authors first speak about PI, then kapton, then double side kapton tape, then of kapton HN on Alumina substrates. The stack structure remains unclear.
2) Measure and report the value of the electrical resistance of a single LIG strip, i.e. the surface resistance of the LIG film. This value should be correlated to the impedance plot in figure 8, at low frequencies.
Response: The authors are very thankful to the Editor and Reviewers for the swift review and insightful comments about this manuscript. The constructive comments will definitely enhance the quality of the manuscript. We have revised the manuscript in accordance with the comments (All the changes are highlighted in yellow in the revised manuscript file).
Please find below the replies and corrections carried in accordance to each of the specific comments:
- Thank you for this suggestion regarding the addition of a scheme of the "polymer sandwich" to better explain the device structure. In order to avoid any misunderstanding, we would like to express firstly that this work employed a commercially available polyimide-based polymer as substrate in order to produce the laser-induced Graphene layers. If it is meant with this suggestion to illustrate the sandwich-structure of this polymer substrate, we have no access to the polymer-sandwich scheme of this commercial product that was supplied by DuPont. If it concerns the sandwich scheme of a micro-supercapacitor (MSC), there are many review articles available in literature that present such illustrations on micro-supercapacitor fabrication. Our main emphasis on this article has been on the microstructure and formation of interdigital Graphene electrodes produced by laser irradiation and their effect on yielding better micro-supercapacitor properties. The structure of the LIG-MSC electrodes together with polymer gel electrolyte has been shown in Figure 2. For clear understanding, we have also modified the section 2.2 in the revised manuscript in page 3 as given below:
2.2. Fabrication of CO2-Laser-induced interdigital structured graphene electrodes
A commercial polyimide (PI)-based Kapton® HN foil (hereafter given as Kapton Foil) with a thickness of 125 μm was used as a flexible substrate and as the carbon precursor for the production of Graphene layers. Prior to the laser irradiation with CO2-Laser, the Kapton foil surface was pretreated with a 1M KOH solution. For that, the KOH solution was drop-casted onto the Kapton foil surface which was then dried in a vacuum desiccator for overnight under reduced pressure. Following this, the Kapton foil was attached onto an aluminum oxide ceramic plate by means of a double-sided Kapton tape to fix and stabilize the Kapton foil for further irradiation processing with CO2-Laser (supplied by VE-VOR, Germany). The irradiated Kapton foil was washed with isopropanol solution and dried in a vacuum desiccator over silica gel pellets at reduced pressure. After this pretreatment and stabilization, the Kapton foil was scanned in a continuous wave (CW) CO2-Laser engraving machine with λ = 10.6 µm in the geometry shown in Figure 1. The dimension of such fabricated MSC-devices was 24 × 10.5 mm. The interdigital Graphene electrodes displayed a hair comb formation which had 12 branches of 7.5 × 0.8 mm equally distributed on each bar of the comb (Figure S1). The laser scanning speed was 8 mm s-1 and had a laser power of 15 % of the maximum power being equivalent to 6.0 W. The distance between the lens and substrate was 55 mm. The manufacturing of the cells was performed at room temperature and in ambient air.
- Authors thank to the Reviewer for this valuable suggestion regarding measurement of the electrical resistance of a single LIG strip, i.e. the surface resistance of the LIG film and correlate this value to the impedance measurement at low frequencies. We will definitely include this suggestion in our next characterization work on LIG-based micro-supercapacitor studies.
Reviewer 2 Report
This paper reported a flexible micro-supercapacitor prepared by the laser-induced graphene electrode and investigated the electrochemical performance. The laser-induced preparation of graphene electrodes has been widely reported, and the novelty of this work has not been well demonstrated. In addition, in the introduction part, the purpose of this work and the summary of the current researches on the graphene-based microsupercapacitors are not well clarified. Another main problem of this work is the preparation process was not well-tuned to achieve the best performance. Thus, based on the above points, I do not think this paper can be accepted.
Author Response
Reviewer 2.
Reviewer’s comments:
This paper reported a flexible micro-supercapacitor prepared by the laser-induced graphene electrode and investigated the electrochemical performance. The laser-induced preparation of graphene electrodes has been widely reported, and the novelty of this work has not been well demonstrated. In addition, in the introduction part, the purpose of this work and the summary of the current researches on the graphene-based microsupercapacitors are not well clarified. Another main problem of this work is the preparation process was not well-tuned to achieve the best performance. Thus, based on the above points, I do not think this paper can be accepted.
Response: The authors are very thankful to the Editor and Reviewers for the swift review and insightful comments about this manuscript. We are sure that the constructive comments will definitely enhance the quality of the manuscript and make it distinguished from the mass of similar papers. We have revised the manuscript in the light of the given comments and hope to find the approval of the Reviewer 2. (All the changes are highlighted in yellow in the revised manuscript file). In fact, this entire research work exhibits the combination of various novel approaches, as it was indicated by the other two Reviewers. These cover the LIG-synthesis approach, polymer gel embedded ionic liquid electrolyte preparation, supercapacitor characterization studies and explanations regarding morphology of electrodes and MSC properties. Nevertheless, to meet the requirements regarding the novelty relevant explicit statements, we have modified the introduction part of the revised manuscript in page 2 to provide the purpose of this work more precisely and explicitly. Moreover, for clarity, we have also modified the section 2.2 in revised manuscript on page 3. With that, we are hoping to find the approval of the Reviewer 2 to accept the revised paper for consideration.
Page 2: The emphasis of this research work is on the fabrication of LIG-based EDLC electrodes at ambient temperature for use in flexible and scalable micro-supercapacitor. This work has the ambition to display the significant impact factors for improved MSC performance through embedded ionic liquid electrolyte and optimized ion transport pathways via the increased electrode porosity yielding a wider potential window.
Page 3:
2.2. Fabrication of CO2-Laser-induced interdigital structured graphene electrodes
A commercial polyimide (PI)-based Kapton® HN foil (hereafter given as Kapton Foil) with a thickness of 125 μm was used as a flexible substrate and as the carbon precursor for the production of Graphene layers. Prior to the laser irradiation with CO2-Laser, the Kapton foil surface was pretreated with a 1M KOH solution. For that, the KOH solution was drop-casted onto the Kapton foil surface which was then dried in a vacuum desiccator for overnight under reduced pressure. Following this, the Kapton foil was attached onto an aluminum oxide ceramic plates by means of a double-sided Kapton tape to fix and stabilize the Kapton foil for further irradiation processing with CO2-Laser (supplied by VE-VOR, Germany). The irradiated Kapton foil was then washed with isopropanol solution and dried in a vacuum desiccator over silica gel pellets at reduced pressure. After this pretreatment and stabilization, the Kapton foil was scanned in a continuous wave (CW) CO2-Laser engraving machine with λ = 10.6 µm in the geometry shown in Figure 1. The dimensions of such fabricated MSC-devices were 24 × 10.5 mm. The interdigital Graphene electrodes displayed in a hair comb formation which had 12 branches of 7.5 × 0.8 mm equally distributed on each bar of the comb (Figure S1). The laser scanning speed was 8 mm s-1 and had a laser power of 15 % of the maximum power being equivalent to 6.0 W. The distance between the lens and substrate was 55 mm. The manufacturing of the cells was performed at room temperature and in ambient air.
Reviewer 3 Report
The authors report the fabrication of structured graphene through Laser-induced method and employed as electrodes for micro-supercapacitors encompassing a polymer-gel electrolyte. The as-obtained electrode was investigated through SEM, EDX, and XRD tools to verify the physico-chemical properties. Further, the developed Laser-induced graphene electrodes were tested in micro-supercapacitor configuration to show interesting electrochemical performances. The overall motivation of this work is well substantiated through the results and discussion. Considering the methodology of graphene preparation and its application to one of pressing demands showing interesting results, i.e., energy storage and technology, this work could be accepted for publication after addressing the points below.
- Abstract contains too much of introduction, suggest briefing the motivation of this work.
- What is thickness of the graphene electrodes?’
- What is the conversion or yield of graphene from this source and method?
- Raman spectra should be provided to better understand the D and G band signatures.
- How to compare the results with other electrolyte system. Suggest adding few studies in Table 1.
- For clarity, EIS plot depicting the high-frequency region be enlarged and provide as inset.
- Unless the polymer-gel electrolyte is not a novel, reference should be provided.

Author Response
Reviewer 3.
Reviewer’s comments:
The authors report the fabrication of structured graphene through Laser-induced method and employed as electrodes for micro-supercapacitors encompassing a polymer-gel electrolyte. The as-obtained electrode was investigated through SEM, EDX, and XRD tools to verify the physico-chemical properties. Further, the developed Laser-induced graphene electrodes were tested in micro-supercapacitor configuration to show interesting electrochemical performances. The overall motivation of this work is well substantiated through the results and discussion. Considering the methodology of graphene preparation and its application to one of pressing demands showing interesting results, i.e., energy storage and technology, this work could be accepted for publication after addressing the points below.
Response: The authors are very thankful to the Editor and Reviewers for the swift review and insightful comments about this manuscript. The constructive comments will definitely enhance the quality of our manuscript. We have revised the manuscript in accordance with the given comments (All the changes are highlighted in yellow in the revised manuscript file) and present below our replies to each specific comment.
- Abstract contains too much of introduction, suggest briefing the motivation of this work.
Response: Thank you for this comment and suggestion. We have modified the introduction part in revised manuscript and included few sentences about the motivation of this work. Please see the revised manuscript in page 1.
Abstract: The rapid developing demand on lightweight portable electronics accelerate the advance research on self-powered microsystems (SPMs) for peak power energy storage (ESs). In recent years, there has been, in this regard, a huge research interest in micro-supercapacitors for microelectronics application over micro-batteries due to their advantages of fast charge-discharge rate, high power density and long cycle-life. In this work, the optimization and fabrication of micro-supercapacitors (MSCs) by means of laser-induced interdigital structured graphene electrodes (LIG) has been reported. The flexible and scalable MSCs are fabricated by CO2-laser structuring of Polyimide-based Kapton ® HN foils at ambient temperature yielding interdigital LIG-electrodes and using polymer gel electrolyte (PGE) produced by Polypropylene carbonate (PPC) embedded ionic liquid of 1-Ethyl-3-methyl-imidazolium-trifluoromethansulphonate [EMIM][OTf]. This MSC exhibits a wide stable potential window up to 2.0 V, offering an areal capacitance of 1.75 mF/cm2 at a scan rate of 5.0 mV/s resulting in an energy density (Ea) of 0.256 mWh/cm2 @ 0.03 mA/cm2 and power density (Pa) of 0.11 mW/cm2 @0.1 mA/cm2. Overall electrochemical performance of this LIG/PGE-MSC is rounded with a good cyclic stability up to 10000 cycles demonstrating its potential in terms of peak energy storage ability compared to the current thin film micro-supercapacitors.
- What is thickness of the graphene electrodes?’
Response: Thank you for this comment. We have mentioned the thickness of the graphene electrode in the revised manuscript. Please see the revised manuscript in page 6.
The porous LIG resulted through the conversion of PI-foil yielding a thickness of varying between 50 and 94 µm.
- What is the conversion or yield of graphene from this source and method?
Response: Thank you for this comment. In this work, the sample of polyimide (PI)-based Kapton foil were irradiated through continuous wave (CW) CO2-Laser scanning to create graphene.
The conversion or yield of graphene layer thickness obtained through laser irradiation depends mainly on the laser power as well as surface treatment of the PI-foil. This is displayed by applying different laser powers and use of untreated and KOH surface treated foils. By starting with an untreated (No pretreatment-No PT) PI-foil of 125 µm thickness and a laser power of 4.8 W, approx. 18 – 30 µm polymer was converted into porous carbon-based layer. The thickness of this resulting porous layer varied depending on the applied process parameters between 50 – 94 µm. The increase of laser power from 4.8 W to 5.6 W has also increased the converted PI-foil thickness from 18 to 31 µm. A further increase of the laser power to 6 W, the penetration depth of the laser decreased to 20 µm, indicating a laser power of 5.6 W yields maximum conversion of the untreated PI-foil surface. On the other hand, a surface pre-treatment of the PI-foil using KOH (KOH-PT) yields a significant growth of the LIG layer at higher laser powers (5.2 – 6 W). The maximum LIG layer thickness (94 µm) has been achieved with 6 W and on the KOH pretreated PI-foils. These resulted also in better layer morphologies as well as higher yields. While, the PI surfaces irradiated without any pretreatment deviated slightly from a linear increase. However, over 6 W exceeding increase of laser power (e.g. 6.4 W and 8 W) on KOH-PT foil resulted in a largely deviating layer thickness and the wave-like surface topology
- Raman spectra should be provided to better understand the D and G band signatures.
Response: The authors are extremely grateful to the Reviewer for giving them this valuable advice to measure the Raman spectra to better understand the D and G band signatures. We had some preliminary Raman spectroscopic measurements with the help of a partner institute, in order to identify the quality of the layers produced by different surface treatments. These were also compared with the spectra of glassy carbon and single layer Graphene. We are disclosing below some graphs from these measurements for consideration of the Reviewer. In our opinion, a more detailed study is required in this aspect. Unfortunately, the time given for revision is limited to do so. Our next planned studies will definitely involve more intensive Raman spectroscopy analysis on LIG-based micro-supercapacitor electrode manufacturing.
However, if it is seen by the Reviewer as necessary, we can add the below text for Raman Spectroscopy Measurements:
For Raman spectroscopy, a scraped LIG sample was produced by scanning a 20 x 20 µm raster with a 5 µm laser spot. The Raman spectroscopy of this scraped LIG sample was recorded at room temperature on a Bruker Senterra Raman spectrometer with λ = 532 nm by Dr. Shujie You at the Faculty of Engineering and Mathematics at Luleå University of Technology in Sweden. The maximum power for laser excitation was below 0.02mW to avoid any possible heating and recrystallization of the sample. For sample focusing a 50× object (Olympus LMPlan N) with a working distance of 10.6 mm was used.
Figure 1. Raman spectra recorded from different spots of LIG sample with 532 nm excitation. The spectra were renormalized and stacked for the clear view. The spectra arranged in groups based on the similarities, name with Type I, II and III.
The three spectra in Type I present the defect related band (D-band, center at 1349 cm-1), graphite-like band (G-band, center at 1587 cm-1) and a 2D-band (center at 2695cm-1) due to two-phonon vibration.[1] It is well known that the profile of 2D band and the relative intensity ratio of I2D/IG are often employed as the criteria to identify the number of layers of Graphene. The symmetric profile and relatively high I2D/IG ratio in spectra in Type I (Figure 1) indicates that these black and the magenta spectra are likely from graphene-like structure. The broad band at around 1350 cm-1 is usually assigned to defect related vibration, it is also found that the intensity of D band is higher at the edge of graphene structure and thus is used to estimate the crystallite size as in the following equation:[2]
Besides, the shoulder of G-band, at 1620 cm-1, which is not common in graphite/bulk materials but often present in nanoribbons,[3][4] is also considered to be defect/size related.
- How to compare the results with other electrolyte system. Suggest adding few studies in Table 1
Response: Thank you for this comment and suggestion. The best of our knowledge there are very few numbers of publication available in literature on polymer embedded ionic liquid-based electrolytes for micro-supercapacitor application in this wide range of potential windows (up to 2 V). Nevertheless, we have included the most important and relevant ones to yield a comparison with our results in Table 1.
- For clarity, EIS plot depicting the high-frequency region be enlarged and provide as inset.
Response: Thank you for this suggestion. For better clarity, we have included the EIS plot depicting the high-frequency region and included in revised Supplementary File. Please see the revised Supplementary File in Figure S3.
- Unless the polymer-gel electrolyte is not a novel, reference should be provided.
Response: Thank you for this comment. The polymer gel electrolyte is produced in our laboratories using PPC polymer matrix and ionic liquid [1-Ethyl-3-methyl-imidazolium-trifluoromethansulphonate [EMIM][OTf] (99%, M=260.24 g mol-1, ?=1.39 g cm-3] provided by Company IoLiTec (Heilbronn, Germany)]. This polymer gel electrolyte with another ionic liquid [1-Ethyl-3-methylimidazolium bis(trifluoromethylsulfonyl)imide [EMIM][TFSI] (M=391.31 g mol-1 =1.52 g cm-3, 99.5% purity) also provided by Company IoLiTec (Heilbronn, Germany) is published previously in one of our papers.
The reference for which has been included in revised manuscript (Ref. 18). Please see the revised manuscript on page 3.
[1] a Ferrari, “Raman Spectroscopy of Graphene and Graphite: Disorder, Electron–Phonon Coupling, Doping and Nonadiabatic Effects,” Solid State Communications 143, no. 1–2 (July 2007): 47–57, https://doi.org/10.1016/j.ssc.2007.03.052.
[2] M. A. Pimenta et al., “Studying Disorder in Graphite-Based Systems by Raman Spectroscopy,” Physical Chemistry Chemical Physics 9, no. 11 (2007): 1276–91, https://doi.org/10.1039/b613962k.
[3] Pablo Solís-Fernández et al., “Tunable Doping of Graphene Nanoribbon Arrays by Chemical Functionalization,” Nanoscale 7, no. 8 (2015): 3572–80, https://doi.org/10.1039/c4nr07007k.
[4] Sunmin Ryu et al., “Raman Spectroscopy of Lithographically Patterned Graphene Nanoribbons,” ACS Nano 5, no. 5 (May 24, 2011): 4123–30, https://doi.org/10.1021/nn200799y.

Round 2
Reviewer 2 Report
no more comments